# Is Melatonin the Cornucopia of the 21st Century?

**DOI:** 10.3390/antiox9111088

**Published:** 2020-11-05

**Authors:** Nadia Ferlazzo, Giulia Andolina, Attilio Cannata, Maria Giovanna Costanzo, Valentina Rizzo, Monica Currò, Riccardo Ientile, Daniela Caccamo

**Affiliations:** Department of Biomedical Sciences, Dental Sciences, and Morpho-Functional Imaging, Polyclinic Hospital University, Via C. Valeria 1, 98125 Messina, Italy; nferlazzo@unime.it (N.F.); giulia.andolina90@virgilio.it (G.A.); attiliocannata@gmail.com (A.C.); mgc90@hotmail.it (M.G.C.); rizzo_valentina@hotmail.com (V.R.); moncurro@unime.it (M.C.); ientile@unime.it (R.I.)

**Keywords:** sleep–wake cycle regulation, free radical scavenging, anti-inflammatory action, immunomodulation, bone mass protection, fertility amelioration, anti-obesogenic properties, cardiovascular protection, anti-tumoral activity, neuroprotection

## Abstract

Melatonin, an indoleamine hormone produced and secreted at night by pinealocytes and extra-pineal cells, plays an important role in timing circadian rhythms (24-h internal clock) and regulating the sleep/wake cycle in humans. However, in recent years melatonin has gained much attention mainly because of its demonstrated powerful lipophilic antioxidant and free radical scavenging action. Melatonin has been proven to be twice as active as vitamin E, believed to be the most effective lipophilic antioxidant. Melatonin-induced signal transduction through melatonin receptors promotes the expression of antioxidant enzymes as well as inflammation-related genes. Melatonin also exerts an immunomodulatory action through the stimulation of high-affinity receptors expressed in immunocompetent cells. Here, we reviewed the efficacy, safety and side effects of melatonin supplementation in treating oxidative stress- and/or inflammation-related disorders, such as obesity, cardiovascular diseases, immune disorders, infectious diseases, cancer, neurodegenerative diseases, as well as osteoporosis and infertility.

## 1. Introduction

Melatonin is an indoleamine which in mammals is mainly produced by the pineal gland during the night, and is regulated by the hypothalamic suprachiasmatic nuclei and inhibited by light. However, several studies have also shown extra-pineal melatonin synthesis, that is not regulated by circadian cycles, in the gastrointestinal tract, ovaries, lymphocytes, macrophages, retina, and skin [1]. The extra-pineal melatonin production exerts a paracrine or autocrine effect, that superimposes on the neuroendocrine hormone response, mainly working as a local antioxidant [1,2].

Melatonin derives from tryptophan, which is converted to serotonin following hydroxylation and decarboxylation reactions. Serotonin is converted to melatonin in two sequential reactions catalyzed by arylalkylamine-*N*-acetyl transferase (AANAT) and hydroxy-indole-*O*-methyltransferase (HIOMT), also known, respectively, as serotonin *N*-acetyl-transferase (SNAT) and *N*-acetylserotonin *O*-methyltransferase (ASMT), with *N*-acetyl serotonin as intermediate product. Several studies have proposed that mitochondria are the primary sites of melatonin synthesis due to the presence of melatonin-forming enzymes, SNAT and ASMT, and the chaperone protein 14-3-3, that prevents SNAT degradation and increases SNAT affinity for serotonin [3] (Figure 1).

The lack of SNAT enzyme and the poor availability of tryptophan are the two limiting factors of the melatonin synthesis pathway.

The half-life of melatonin ranges from 20 to 40 min, and after its secretion melatonin immediately diffuses into the blood and the cerebrospinal fluid (CSF). Due to its lipophilic and hydrophilic properties, it diffuses easily through cell membranes and can be detected in other body fluids such as saliva, milk, sperm, amniotic fluid, or as 6-sulfatoxymelatonin (aMT6), the primary melatonin metabolite, in urine. The distribution of melatonin in the body is not homogenous. In CSF, the concentration is higher than that in blood; however, it is generally accepted that changes in melatonin blood levels, ranging from few pg/mL during the day to 50–100 pg/mL at night, represent the variations of levels throughout the body [4]. This physiological range of concentrations is pharmacologically reached by the oral administration of 0.1–0.3 mg melatonin. However, studies have investigated also higher doses (up to 10 mg) that are considered supra-physiological, reporting no toxic effects. Moreover, the pharmacokinetic properties of melatonin preparations can vary and affect the bioavailability: 1 to 10 mg can raise plasma melatonin levels from 3- to 60-fold its normal peak [5].

Light and photoperiod changes are the key regulators of melatonin synthesis, so that the seasonal changes of temperature and photoperiod significantly affect melatonin production in human body [6].

In recent decades, there has been increased interest in the measurement of melatonin in biological fluid as a marker of circadian phase or to understand the physiological role of melatonin. The measurement of blood melatonin concentrations is often limited due to the difficulty of collecting multiple samples over the 24 h; therefore, non-invasive analytical techniques to measure melatonin in saliva or aMT6 in urine have been developed. The determination of salivary melatonin is preferred, although the levels are several folds lower (up to 10 fold) than those found in plasma. In addition, due to its low concentration and the coexistence of several other similar endogenous compounds, there has been increased attention to set up highly specific and sensitive methods, such as physico-chemical (e.g., liquid chromatography (LC), gas-LC (GLC)-mass spectrometry, and LC-mass spectrometry) and immunological methods e.g., radioimmunoassay (RIA) and enzyme-linked immunosorbent assays (ELISA), although with some limitations. In particular, cross-reactions and nonspecific binding must be considered during RIAs and ELISAs procedures, that are the most commonly used methods for the determination of melatonin in blood or saliva. In addition, given the low melatonin concentrations in body fluids, adequate extraction methods, as well as correct handling and preservation of samples from light, high temperature and oxidation, due to high oxygen levels, are necessary for an accurate determination of melatonin levels by LC [4].

Being an amphipathic molecule, once produced melatonin is able to cross cell membranes and reach different body areas, where it exerts various biological effects and functions not only through the binding with receptors, but also through its direct interaction with other molecules [6].

Melatonin binds with high affinity to the membrane-associated receptors MT1 and MT2, that are typically coupled to G proteins. The activation of either MT1 or MT2 leads to inhibition of adenylate cyclase (AC)/cyclic adenosine monophosphate (cAMP)/protein kinase A (PKA)/cAMP response element-binding protein (CREB) pathway, or guanylate cyclase (GC)/cyclic guanosine monophosphate (cGMP)/protein kinase G (PKG) pathway, respectively, as well as to activation of phospholipase C (PLC) pathway, with subsequent increase of inositol triphosphate (IP3) and 1, 2-diacylglycerol (DAG) (Figure 1). A third site, named MT3, binding melatonin with low affinity, has been characterized as the cytosolic enzyme quinone reductase 2 (QR2). In addition, melatonin binds to nuclear receptors, such as retinoid-related orphan receptors (RZRα/ROR, RZRβ/ROR) [7] (Figure 1). Notably, ROR receptors regulate the biological clock circuitry and play a key role in the integration of circadian outputs and metabolic processes [8].

Signaling pathways activated by MT receptors are highly cell- and tissue-dependent, and also dependent on different proteins forming heterodimer complexes with MT receptors [7]. These features are fundamental for melatonin pleiotropic effects in human body, which can be regarded as both regulatory of key metabolic processes and protective against various disorders (Figure 2).

This work will review melatonin role in the regulation of sleep–wake cycle, bone metabolism, as well as fertility and reproduction, and its protective effects against oxidative stress, abnormal immune activation and/or inflammation, obesity, cardiovascular disorders, cancer development, and neurodegeneration. Melatonin ability to ameliorate mood disorders, and prolong physical performance as well as limit skeletal muscle frailty, will not be discussed here, since they have recently been reviewed in exhaustive way [9,10].

## 2. Regulatory Effects of Melatonin

### 2.1. Modulation of Sleep–Wake Cycle

The sleep–wake rhythm, one of the most important neurophysiological processes dependent on central nervous system (CNS) regulation, is composed by two fundamental phases interacting one with each other: one that controls the circadian rhythm (process C) and one that regulates the various phases of sleep (process S). Process C, guided by suprachiasmatic nucleus (SCN) of anterior hypothalamus, is the pacemaker that regulates the circadian rhythms of rapid eye movement (REM) sleep, wakefulness and many other physiological rhythms, and interacts with process S that increases during wakefulness and decreases during sleep [11]. One of the most important regulators of sleep–wake rhythm is melatonin, that interacts with MT1 and MT2 receptors in SCN of hypothalamus and retina, promoting sleep and inhibiting wake-promoting signals [12]. Melatonin secretion reaches a maximum peak between 2:00 and 4:00 a.m., and then gradually decreases with the exposure to early morning bright light. Melatonin secretion adapts to the length of the night, and is responsible for the modulation of daily light-dark cycle. Melatonin production is influenced by invasion of sleep–wake cycle, such as undergoing excessive brightness just before bedtime, being subjected to stressful situations, and jet-lag syndrome. Several studies demonstrated that blue light of different electronic devices (computers, smartphones, tablets) reduces melatonin release, and, as a consequence, also sleep duration and quality [13,14,15,16,17]. Two hours of continuous use of tablet and smartphone in the evening reduces melatonin production by 22% in 20-year-old subjects, that showed a propensity to sleep less and fall asleep with difficulty compared to the average age [18].

Forced sleep deprivation negatively influences daily life and affects individual health. The misalignment of the sleep–wake rhythm is common in rotating shift work, which also includes the night shift work. This leads professionals to complain of daytime fatigue and greater difficulty in falling asleep at different times of the day [19]. In modern society, shift work has become an essential part of the work system, but it has been shown to be associated with a higher incidence of tumors and various pathological conditions, including sleep disorders, as well as cardiovascular and gastrointestinal disorders [20].

Among dietary supplements, synthetic melatonin, commercially available in different formulations, is considered a first-line pharmacologic therapy for treatment of insomnia by the American Academy of Family Physicians [21].

It has been shown that night workers should sleep during the day in bright light conditions rather than in low light conditions in order to improve the melatonin secretion and maintain the conventional sleep–wake cycle. Melatonin administration at bedtime during a night shift can improve sleep and increase daytime vigilance in shift workers, and therefore can be a useful strategy to help real night workers adapt to work shifts at night [22].

Altered melatonin secretion has been associated with significant reduction in sleep efficiency and continuity typical of elderly individuals [23]. Some studies have suggested that aging is actually a syndrome resulting from melatonin deficiency [24,25]. The total melatonin production throughout 24 h seems not to change in healthy elderly, but the night peak concentration decreases [26]. Thus, the treatment of elderly people with melatonin may give an improvement in sleep quality, next day alertness and life quality, as previously demonstrated [27]. In this regard, a prolonged-release melatonin (Circadin) has been approved by the European Medicines Agency for short-term treatment of primary insomnia characterized by poor sleep quality in patients aged 55 years or older [28].

A phase shift in melatonin secretion also occurs in flight passengers crossing the hour change line and in patients with delayed sleep phase syndrome. Moreover, low melatonin levels have been found in most individuals with autism spectrum disorder (ASD) [29]. In particular, it has been highlighted that abnormal melatonin metabolism can be one cause of sleep disorders associated with ASD [30]. Recently, a Randomized Placebo-Controlled Trial (RPCT) has been performed to evaluate the efficacy of a pediatric-appropriate, prolonged-release melatonin minitablet (PedPRM) [31] for insomnia treatment in children and adolescents with ASD, with or without attention-deficit/hyperactivity disorder comorbidity, and neurogenetic disorders [32]. Results demonstrated that PedPRM (once daily 2- or 5-mg dose) increased total sleep time, reduced sleep latency, and improved longest continuous sleep period throughout 13 weeks of treatment. No effects on seizures were reported, while mild adverse events, such as somnolence, headache, and fatigue, were recorded. No evidence of tolerance development was found [32].

Despite several encouraging results, a recent analysis of the data in the literature evaluating the efficacy of melatonin in primary and comorbid insomnia disorders evidenced a statistically significant improvement in sleep latency and total sleep time. However, a lack of consensus still remains regarding its clinically meaningful benefits due to disparate measurements of type, definition and interpretation of outcome [33]. In addition, doses used in studies varied from 0.1 to 10 mg [21], only short term studies have been carried out and no evidence have been provided for melatonin effectiveness or safety beyond 16 weeks of treatment [34].

The disruption of the circadian rhythm, related to melatonin deficiency, and the resulting interrupted sleep have both been linked to metabolic and cardiovascular diseases, cancer risk, neurological diseases, and mood disorders [35]. A deeper understanding of the relationship between cellular metabolism and sleep/wake cycle control will be helpful to define effective strategies for disease prevention and good health recovery [36].

### 2.2. Melatonin Role in Fertility and Reproduction

Classified as an anti-estrogen hormone, melatonin has also been shown to influence the reproductive sphere of both women and men [37,38].

Melatonin regulates the production of prolactin (PRL) in lactotropic cells, as well as follicle-stimulating hormone (FSH) and luteinizing hormone (LH) in gonadotropic cells. The reduction of FSH and LH levels protects the body from the risk of early entry into puberty [39]. This occurs as a result of the decrease of cAMP-dependent gonadotropin release hormone (GnRH) levels, following the activation of melatonin receptors, associated with a reduction in the calcium influx [40]. Furthermore, MT1 and MT2 play a key role during reproduction since melatonin regulates p38 activation which can positively influence the embryonic implant [41]. In women, it can be also synthesized by follicular cells, where it is involved in ovulation, and in placenta tissue homeostasis [37]. Furthermore, as a result of its anti-estrogenic properties, melatonin has also been shown to block the production of estradiol [42].

Melatonin has positive effects in gynecological disorders, such as polycystic ovary syndrome (PCOS), premature ovarian failure, and oophoritis, by reducing follicular cell death due to its anti-apoptotic activity [43]. PCOS, a widespread endocrine disorder affecting about 20% of women within reproductive age, is characterized by hyperandrogenism, obesity, menstrual irregularity, and anovulatory infertility. Melatonin treatment for six months was able to restore menstrual cyclicity in women (*n* = 40) with PCOS, by significantly decreasing androgens and anti-Mullerian hormone serum levels and significantly raising FSH levels [44]. Higher levels of melatonin have been found in women affected by PCOS, likely as a compensative response resulting from body attempts to neutralize high reactive oxygen species (ROS) amounts produced in this disorder [45]. Results from a randomized, double-blinded, placebo-controlled clinical trial carried out in women with PCOS (*n* = 56, age 18–40 years) receiving melatonin for twelve weeks, demonstrated a decrease in hirsutism and testosterone, as well as in oxidative stress markers, such as malondialdehyde, high sensitivity C reactive protein (hs-CRP), and gene expression of inflammatory cytokines interleukin-1 (IL-1) and tumor necrosis factor-α (TNF-α), were reported, together with the elevation of reduced glutathione (GSH) levels and total antioxidant capacity [46]. Melatonin has been shown to improve oocyte development potential thanks to its anti-inflammatory and antioxidant effects mediated by a significant reduction of inducible nitric oxide synthase (iNOS) and nitric oxide (NO) levels in luteinized granulosa cells, and increased levels of mRNAs for the antioxidant transcription factor nuclear factor erythroid 2-related factor 2 (Nrf-2) and its downstream target heme oxygenase-1 (HO-1) [47]. Notably, the intake of melatonin associated with myo-inositol has been shown to synergistically improve oocyte and embryo quality, clinical pregnancy and implantation rates [48,49].

It has been observed that some cases of PCOS are associated with hyperinsulinemia. Interestingly, melatonin treatment displayed beneficial effects on insulin levels and insulin resistance index (HOMA-IR), as well as on cholesterol levels, and peroxisome proliferator-activated receptor -γ (PPAR-γ) and low-density lipoprotein receptor (LDLR) expression among women with PCOS; moreover, an improvement of mental health parameters was also observed [50].

A phase II, double-blind RPCT showed that an 8-week melatonin treatment in women suffering from endometriosis (*n* = 40, age 18–45 years) significantly reduced daily pain (Δ = −39.8%) and dysmenorrhea (Δ = −38.01%), and also reduced the concentrations of brain derived neurotrophic factor (BDNF) by a mechanism distinct from that one alleviating pain [51]. Moreover, experimental studies carried out on cultured normal and endometriotic endometrium epithelial cells isolated from women of reproductive age (*n* = 6 per each group), showed that melatonin was able to reduce cell proliferation, invasion, and migration, as well as the levels of epithelial-mesenchymal transition markers, by blocking at once the estradiol and Notch signaling pathways [52]. However, high doses of melatonin were toxic to follicular cells, and induced amenorrhea by decreasing the secretion of gonadotropins and PRL [37].

The administration of melatonin induces an increase in prostaglandins and cytokines and a consequent reduction of ROS levels, the elevation of which is positively associated with the increase of follicular aging. As a consequence, oocytes are protected from oxidative stress. Some studies demonstrated that the concentration of melatonin in the follicular pre-ovulatory fluid is about three times greater than in serum [53], similarly to what happens during pregnancy [54]. Thus, the treatment of melatonin in poorly fertile women reduces oxidative stress and the concentration of estradiol, both higher than in fertile women, and, in parallel, increases both the quality of the oocytes and their intrafollicular concentration, and consequently the fertility and probability of pregnancy [55,56,57].

A non-regular sleep–wake cycle can negatively influence menstrual regularity and the probability of embryonic implantation, with consequent repercussions on pregnancy [58]. Melatonin production increases during pregnancy as well the levels of placental melatonin increase as the placenta grows. However, although the pineal gland of a fetus can produce melatonin, the circadian effects are determined by maternal melatonin, which is important for the morphological and functional development of SCN and pineal gland and other rhythmic body systems [59]. The absence of melatonin production in pregnant women has been shown to occur with maternal obesity and metabolic syndrome as well as in pregnancies complicated by preeclampsia, chronic placental insufficiency, or night work. Melatonin deficiency will disrupt the development of SCN and circadian rhythm, with consequences for metabolism [60]. The lack of a circadian melatonin rhythm in obese mothers is associated with the absence of circadian melatonin rhythm in their offspring; instead, the offspring of non-obese mothers shows a low but significant circadian melatonin rhythm from the 3rd postnatal day [61]. Melatonin is therefore an important regulator of the mother–placenta–fetus interface. Such data show the importance of the melatonin circadian rhythm over pregnancy [62].

Melatonin also regulates male fertility by modulating the endocrine function of Leydig cells and steroidogenesis in Sertoli cells thanks to its influence on cell proliferation and energy metabolism. A continuous treatment with melatonin has negative feedback on LH, diminishing the production of bound testosterone. On the other hand, too low concentrations of melatonin have been shown to induce a reduction in testicular mass [63].

The pre-treatment with melatonin displayed anti-apoptotic effects through receptor-mediated activation of extracellular signal-regulated kinase (ERK) pathway in human spermatozoa exposed to hydrogen peroxide [64]. The supplementation of melatonin either alone [65] or in combination with myo-inositol [48] has been shown to improve male semen quality. The daily supplementation of melatonin for 45 days of treatment enhanced both urinary and seminal total antioxidant capacity, and consequently reduced oxidative damage caused in sperm DNA [66].

The continuous treatment with melatonin improved the cellular response for erectile dysfunction through the increase in vasodilatory cytokines [67,68]. In an in vivo study, melatonin improved the motility, integrity of the membrane and the potency of rabbit sperm become more fragile due to freeze–thaw-induced oxidative stress [69]. Furthermore, melatonin was able to rescue the penetration capacity of human spermatozoa impaired by mitochondrial dysfunction [38,70].

Most of findings previously reported on melatonin efficacy in fertility disorders have been obtained either in animal studies or in small sample size human studies. Thus, melatonin’s beneficial effects should be further confirmed in large-scale prospective randomized studies.

### 2.3. Melatonin and Bone Metabolism

It has been shown that a decrease in nocturnal melatonin production is associated with a weakening of bone structure and an increased risk of fractures. This risk is usually reduced thanks to the non-pharmacological treatment with vitamin D and calcium. However, pre-clinical and clinical studies suggest that melatonin treatment can effectively reverse bone loss [71].

Experimental data show that melatonin participates in the differentiation process of adult mesenchymal stem cells (MSCs) by activating molecular pathways that involve mitogen-activated protein kinase kinase 1/2 (MEK1/2), ERK1/2, epidermal growth factor (EGF) receptors, matrix metalloproteinase (MMP) and clathrin-mediated endocytosis [72]. Osteogenesis is promoted in a dose-dependent manner in MSCs through suppression of PPARγ, and a variety of mechanisms, that involve increased proliferation or differentiation of osteoblasts mediated by the activation of MEK/ERK1/2, Runt-related transcription factor 2 (RUNX-2), osteocalcin (OCN), bone morphogenetic protein-2 (BMP-2), BMP-4 and wingless-related integration site (Wnt) signal transduction pathways [73,74,75]. Melatonin ameliorates estrogen deficiency-induced osteoporosis and impaired osteogenic differentiation potential by suppressing activation of the nucleotide-binding domain leucine-rich repeat (NLR) and pyrin domain containing receptor 3 (NLRP3) inflammasome via the Wnt/β-catenin pathway [76]. Moreover, it has been shown to induce osteoblastogenesis and inhibit osteoclastogenesis through the regulation of receptor activator of nuclear factor kappa-Β ligand (RANKL), and osteoprotegerin synthesis and release from osteoblasts [77].

These features have been clinically exploited in bone-grafting procedures, in counteracting bone loss due to osteopenia and osteoporosis, and in managing periodontal disease [78]. It has been reported that elderly women benefit from the use of melatonin as a dietary supplement in addition to routine medications, and not used as a main drug [79]. A double-blind, placebo-controlled trial showed that a six-month nightly treatment of women aged 45–54 with melatonin (3 mg, per os) was able to normalize the turnover of bone cells, measured as the trend downward over time toward 1:1 of the ratio between *N*-telopeptide of Type I Collagen and osteocalcin (NTX/OC), and improve physical domain scores, without effects, however, on vasomotor, psychosocial, or sexual Menopause-Specific Quality of Life-Intervention domain scores. Moreover, melatonin reduced menstrual cycling while significantly increasing days between cycles in comparison with placebo (51.2 vs. 24.1 days) [80,81,82,83].

A randomized controlled trial, involving a one-year nightly treatment with melatonin (1 mg, 3 mg) in women aged 56–73 years, demonstrated that melatonin increased femoral neck bone mineral density (BMD) in a dose-dependent manner (0.5% at 1 mg/day; 2.3% at 3 mg/day). Moreover, the 3 mg/day melatonin dose increased trabecular thickness in tibia by 2.2%, and volumetric bone mineral density in the spine by 3.6%, though did not significantly affect BMD at other skeleton sites or bone turnover markers; however, 24-hr urinary calcium excretion was significantly decreased by 12.2% [80].

Melatonin is able to preserve the extracellular matrix (ECM) content of collagen II, aggrecan and sox-9 by inhibiting the expression of MMP-13 and of a disintegrin and metalloproteinase with thrombospondin motifs 5 (ADAMTS-5), involved in matrix degradation. Moreover, melatonin treatment protected cultured nucleus pulposus cells against apoptosis via mitophagy induction and ameliorated intervertebral disc degeneration in a puncture-induced rat model [82].

Recently, a composite adhesive hydrogel system (GelMA-DOPA@MT), releasing melatonin in a sustained way, has been tested around bone implant with the aim of addressing implant loosening in patients with osteoporosis. Melatonin was able to reduce apoptosis in osteoblasts around the implant, and increase bone mass around the implant [83].

## 3. Protective Effects of Melatonin

### 3.1. Melatonin as Antioxidant and Anti-Inflammatory Agent

Oxidative stress is a common feature of various pathological conditions, such as metabolic, degenerative and cardiovascular disorders, as well as cancer. Melatonin has been shown to act as a powerful antioxidant, more effective than vitamin E, and plays a protective role both intracellularly and extracellularly [6]. Melatonin is abundant in mitochondria, where the highest production of ROS takes place during metabolism, and effectively prevents the damage of neighboring molecules by free radicals.

ROS scavenging by melatonin may occur both directly and indirectly. The direct action of melatonin is dependent on the buffering capacity of its aromatic indole ring reacting with ROS or reactive nitrogen species (RNS). These reactions lead to the formation of metabolites which in turn exhibit antioxidant function through a cascade reaction mechanism, with consequent amplified effect [84,85]. For example, the hydroxylation of melatonin on C3 gives rise to the formation of a cyclic compound, the cyclic 3-hydroxymelatonin (C3-OHM), through a reaction that neutralizes the hydroxyl radical. C3-OHM is subsequently excreted in the urine, thus representing an ongoing scavenging index [86]. The metabolite N1-acetyl-N2-formyl-S-methoxykynuramine (AFMK) is formed following the melatonin-mediated neutralization of hydrogen peroxide and singlet oxygen, and is able to buffer radical species, prevent DNA and protein damage and lipid peroxidation, reducing death of cells exposed to hydrogen peroxide. Another metabolite is the N1-acetyl-5-methoxykynuramine (AMK), which is formed by cleavage of AFMK pyrrolic ring and acts by inhibiting NOS in vitro in a dose-response manner. AFMK and AMK display scavenger activities and may represent valid biomarkers of ROS in vivo exposure; however, AFMK demonstrates poor scavenging activity compared to AMK [86]. Moreover, AMK scavenges NO leading to the formation of a stable nitrosation product, modulates mitochondrial metabolism, and interacts with aromatic rings by forming adducts with tyrosine and tryptophan residues that lead to protein modification Two other metabolites, N1-acetyl-5-methoxy-3-nitrokynuramine (AMNK) and 3-acetoamidomethyl-6-methoxycinnolinone (AMMC), are formed when AMK neutralizes RNS [84].

The indirect antioxidant action of melatonin involves melatonin receptors. Various in vitro and in vivo studies have shown that the activation of MT1 and MT2 receptors stimulates the expression and hence the activity of endogenous antioxidant enzymes, among which are superoxide dismutase (SOD), catalase (CAT), glutathione peroxidase (GPx), and glutathione reductase (GRd). Melatonin also protects antioxidant enzymes from oxidative damage, and increases the synthesis of GSH and glucose-6 phosphate dehydrogenase (G6PD), an enzyme fundamental for the first and rate-limiting step of the pentose phosphate pathway, in which NADPH is produced and then used for GSH recycling.

Melatonin also increases antioxidant defenses by epigenetically inducing Nrf2, that binds to antioxidant response elements (AREs) located in the promoter region of genes coding for antioxidant enzymes [87,88].

The beneficial effects of melatonin have been tested in various pathological conditions associated with oxidative stress. In vitro observations have shown that small melatonin concentrations (100 µM) are able to reduce low-density lipoprotein (LDL) oxidation, thus hindering the pathogenesis of atherosclerosis [89]. A therapeutic potential for melatonin has also been shown in experimental models of hepato-pulmonary syndrome. A study on Wistar rats showed a melatonin-induced reduction in vasodilation and pulmonary fibrosis, lipoperoxidation and oxidative stress, as well as improvement of the lung weight/body weight ratio and alteration of PCO_2_ and PO_2_ after intraperitoneal administration of 20 mg/kg for 14 days [90]. Other potential benefits linked to melatonin antioxidant action have been shown in experimental models of Alzheimer’s disease (see paragraph 3.6 for details).

Melatonin supplementation may also be useful in the treatment of thrombotic/hemolytic diseases associated to redox state alterations. In a study on Swiss albino mice treated with 50 µM of hemin (a product of hemoglobin degradation), the administration of 20 mg/kg of melatonin for 3 days helped to reduce the amount of ROS and lipid peroxidation induced by hemin. Melatonin also raised the level of GSH and increased the number of circulating platelets going to hinder ferroptosis caused by ROS, an iron-mediated type of cell death, and platelet activation. It also reduced the amount of inflammatory cytokines IL-6, IL-23, TNF-α, while increased the production of anti-inflammatory IL-10, reversing the effects of hemin [91].

Many other researches showed that melatonin is also an anti-inflammatory molecule [92], and is able to regulate the activation of the immune system by reducing both chronic and acute inflammation [93,94], and modulating the expression of pro- and anti-inflammatory cytokines [95].

Melatonin can exert an anti-inflammatory action through the inhibition of nuclear factor κB (NF-κB), a transcription factor activating the expression of pro-inflammatory cytokines. It has been hypothesized that melatonin reduces the acetylation of p52 NF-κB subunit by inhibiting p300 histone acetyltransferase (HAT) activity, thereby decreasing p52 binding and suppressing lipopolysaccharide (LPS)-induced iNOS and cycloxygenase 2 (COX-2) expression [96]. Recently, in vivo and in vitro analysis showed that melatonin administration blunts NF-κB transcriptional activity through a sirtuin 1 (SIRT1)-dependent NF-κB deacetylation in septic mice. In turn, NF-κB inhibition prevented inflammasome activation by decreasing NLRP3 expression and activity, leading to the inhibition of caspase-1 activity and mature IL-1β production [97].

In zymosan- and carrageenan-induced experimental models of inflammation, it has been shown melatonin attenuates the activation of poly (ADP-ribose) polymerase 1 (PARP-1) [93], that is involved in the regulation of NF-κB binding activity to target genes for pro-inflammatory cytokines [98], thus resulting in a protective effects of cellular viability.

Furthermore, melatonin blunts inflammation in the course of chronic inflammatory diseases by lowering the levels of inflammatory mediators, such as IL-6, IL-8, COX-2, and NOS, with marked improvement in post-surgery outcome.

In addition, melatonin limits the production of other mediators of the inflammatory response, such as chemokines, prostanoids and leukotrienes, adhesion molecules [99,100,101], and CRP [102].

In young and middle-aged adult men suffering from unexplained infertility, daily 3 mg melatonin oral supplementation was able to decrease inflammation- and oxidative stress-related markers levels. Indeed, in testicular biopsies from patients with hormonal treatment the expression levels of COX2, NLRP3, IL-1β and TNFα, as well as SOD1 and CAT, were lower than in biopsies from patients who were not taking hormonal supplementation [103].

A recent meta-analysis concluded that melatonin supplementation could be effective in reducing inflammatory biomarkers, particularly the pro-inflammatory cytokines TNF-α and IL-6 [104].

Overall, data in the literature provide insights into the potential benefits of melatonin on the inflammatory and oxidative status associated to pathological diseases.

### 3.2. Melatonin and Obesity

Oxidative stress in combination with a chronic inflammatory state is also known to characterize the obesity condition. Obesity is a growing health problem in industrialized countries mainly related to unhealthy lifestyle, like a sedentary behavior and a fat-rich diet.

The mechanisms involved in the pathophysiology of obesity are complex. The excessive adiposity is a condition characterized by a severe dysfunction of white adipose tissue (WAT), including the modifications of its endocrine function. The association between obesity and oxidative stress was suggested by the observations that lower levels of anti-oxidant systems are present in obese subjects than normal subjects [105]. Oxidative stress can stimulate the differentiation of preadipocytes into adipocytes. In particular, the oxidative stress induced by hydrogen peroxide determines this differentiation by positively regulating transcriptional activators, such as CCAAT/Enhancer Binding Protein-beta (C/EBP-β) and PPARγ, involved in the differentiation of adipocytes [106]. Obesity is also linked to a condition of chronic inflammation, mediated in particular by the cytokines IL-6 and TNF-α, expressed in adipose tissue [107,108]. In this context, melatonin may play an important beneficial role thanks to its known antioxidant properties and its action as a metabolic regulator [109]. Melatonin, in addition to epigenetically modulating Nrf2, as already mentioned, can exert an anti-inflammatory action through the inhibition of NF-κB and NLRP3 [97]. In the context of amelioration of obesity-related characteristics the pineal hormone has been shown to reduce the secretion of TNF-α and IL-6 by adipocytes, increase high-density lipoprotein (HDL) cholesterol, decrease plasma levels of triglycerides, LDL and very low-density lipoprotein (VLDL) cholesterol, and reduce visceral fat [109]. A recent study has highlighted the ability of melatonin to induce apoptotic death of a preadipocyte cell line by decreasing phosphorylated ERK activation and increasing the activation of caspase-3, 8, and 9. In addition to increasing the expression of Bax, a pro-apoptotic protein, melatonin has also been shown to reduce the expression of the anti-apoptotic protein Bcl-2 [110].

A key role in the pathophysiology of metabolic dysfunctions induced by obesity is played by the disruption of the adipokine secretion pattern [111], that may be associated with differences in adipokine secretion levels between visceral and subcutaneous WAT deposits [112]. Adipokines, such as adiponectin, omentin-1, leptin and resistin, are involved in the regulation of energy production and expenditure. Adiponectin is a hormone strongly involved in the regulation of lipid and glucose metabolism, insulin sensitivity, appetite and energy expenditure, and also exerts anti-inflammatory actions [113]. Adiponectin is produced in higher amounts by subcutaneous WAT than by visceral WAT [114]. A negative correlation was found between body weight and adiponectin, and between body mass index (BMI) and adiponectin, whose concentration decreases in the obese patient and increases during weight loss [113]. Variations in adiponectin concentrations also occurred following melatonin supplementation in overweight patients [115]. Melatonin may influence adiponectin secretion in several ways, including the impact of indoleamine on adiponectin signaling pathways, its antioxidant and anti-inflammatory properties, improvement of mitochondrial function and changes of other adipokine levels [113].

Omentin-1, mainly synthesized by visceral WAT, is another insulin-sensitizing and anti-inflammatory adipokine, and has been found to improve insulin-stimulated glucose uptake in adipocytes [116]. Omentin-1 circulating levels have been shown to be decreased in obesity and inversely correlated with BMI, insulin resistance and metabolic syndrome. Melatonin supplementation increases omentin-1 serum levels in patients on a low-calory diet. A higher nocturnal melatonin secretion was found to be positively associated with a lower prevalence of insulin resistance among healthy young women [117]. This relationship can be explained at least in part by the influence of melatonin on adiponectin and omentin-1 secretion.

This evidence strongly supports the use of melatonin in obesity treatment and in the prevention of obesity complications. Furthermore, the use of melatonin may be a good approach to reduce obesity by targeting brown adipose tissue, an active metabolic tissue capable of converting extra energy into heat. For this reason, recently, melatonin has been proposed to be a slimming agent in humans due to its ability to promote the growth and metabolic activity of brown adipose tissue.

Further evidence of melatonin usefulness as an adjuvant in the treatment of obesity and metabolic syndrome derive from a clinical trial carried out in patients with metabolic syndrome, which showed beneficial effects of a 2-month melatonin treatment (5 mg/die) on dyslipidemia, blood pressure and oxidative stress [118].

The problem of obesity may be effectively addressed by analyzing the increase in childhood obesity and focusing attention on the mother’s condition during pregnancy and the intrauterine period of the fetus, that may contribute to offspring obesity risk [59]. Melatonin plays a key role also in this context. Indeed, circadian genes Clock and BMAL1 (Brain and Muscle ARNT-Like 1), also known as protein aryl hydrocarbon receptor nuclear translocator-like protein 1 (ARNTL), by which melatonin exerts its circadian effects, regulate mitochondrial metabolism, daytime glucose and triglyceride levels, as well as lipid synthesis, adipogenesis and adipose tissue metabolism [59].

### 3.3. Melatonin and Cardiovascular Diseases

In the last two decades, growing evidence showed that the reduction of melatonin is a risk factor for various cardiovascular diseases (CVDs). It has been shown that low nocturnal levels of melatonin are associated with an increased risk of ischemic myocardial injury, hypertension, atherosclerosis, heart failure, and drug-induced myocardial damage [119,120,121,122].

The administration of melatonin shows some protective effects against ischemia/reperfusion (I/R) injury in various organs, including heart, brain, kidneys, intestines and liver [123], although these mechanisms are not yet well elucidated. Experimental studies using either isolated hearts or cardiomyocytes showed that the pre-treatment or addition of melatonin directly to the reperfusion medium reduced the incidence of arrhythmias [124]. In rats, it has been shown that melatonin protects the heart from I/R injury by inhibiting mitochondrial permeability transition pore opening, probably via prevention of cardiolipin peroxidation, also preventing the release of mitochondrial NAD(+) and cytochrome C [125]. Melatonin can protect against I/R damage by activating the signaling of SIRT1 and SIRT3, NAD-dependent protein deacetylases that are highly expressed in heart tissue, but are significantly down-regulated under I/R conditions [126,127]. In a murine model of nfarcted heart, melatonin treatment also reduced the expression of acetylated pro-apoptotic proteins, such as FoxO1, p53, NF-κB, Bax, and increased the expression of the antiapoptotic protein Bcl-2 in adipose-derived mesenchymal stem cells [128].

Low levels of melatonin have been shown to represent a pathophysiological factor in the development of hypertension. It is known that pulmonary hypertension is a disease characterized by high pulmonary arterial pressure, which leads to right ventricular hypertrophy and heart failure. In a rat model, melatonin treatment alleviated right ventricular hypertrophy and dysfunction, and also reduced interstitial fibrosis and oxidative stress [129]. Notably, melatonin supplementation reduces nocturnal hypertension, blood pressure, platelet aggregation and circulating catecholamines [130].

The treatment with melatonin and angiotensin converting enzyme inhibitor was able to alleviate pathological changes occurring in rats exposed to continuous light, such as hypertension, hypertrophy and fibrosis of left ventricle (LV) and increased oxidative stress in LV and aorta [131].

Melatonin secretion and circulating levels are reduced in patients with acute and chronic heart failure (HF) [119,132]. Emerging studies suggest that serum melatonin levels are a useful marker for HF. In particular, the melatonin serum levels negatively correlate with the *N*-terminal pro-brain natriuretic peptide (NT-pro-BNP) levels, a known biomarker of HF [119,133].

Melatonin supplementation has been shown to reduce the number and area of atheromatous plaques by modulating the MAPK pathway [134], and also reverse mitochondrial dysfunction and attenuate left ventricular remodeling as well as apoptosis after acute myocardial infarction [135].

Clinical studies indicated that melatonin is able to attenuate complications of myocardial infarction and reduce ischemia-induced myocardial damage as well as ventricle hypertrophy; moreover, it displays beneficial effects as adjunct to surgical and non-surgical treatments of CVDs [130]. However, a strong and well-documented evidence on efficacy of melatonin as therapeutic approach to treatment of CVDs in humans is still lacking, and has yet to be achieved through randomized clinical trials.

### 3.4. Melatonin Immunomodulatory and Anti-Infective Activity

Melatonin also plays an important role in various immune disorders, such as infections, autoimmunity, and immune-senescence. Notably, melatonin immunomodulatory functions are seasonally dependent, so that seasonal changes in melatonin levels may contribute to seasonally dependent disease states associated with an increased incidence of infectious and neoplastic diseases [136].

Melatonin has been shown to act specifically on MT2 receptors expressed in immunocompetent cells and regulate both cellular and humoral responses as well as innate immunity [137]. Notably, not only immune cells like T-lymphocytes, natural killer cells, eosinophils, and mast cells have melatonin receptors, but emerging evidence revealed that the immune system is one source of extrapineal melatonin [137]. Several studies have shown that this indoleamine has the ability to influence the differentiation and trafficking of immune cells (macrophages, natural killer cells, lymphocytes) in response to different events, since these cells follow a circadian rhythm that is influenced not only by melatonin, but also from cortisol and chemokines [138]. Melatonin is able to activate human T helper (Th1) lymphocytes by increasing the in vitro production of IL-2 and IFN-γ, and enhances IL-6 production by peripheral blood mononuclear cells (PBMCs). Additionally, increasing the production of IL-12 enhances natural killer cell (NK) activity [137].

Moreover, tryptophan and its metabolites at intestinal level have been reported to have a beneficial action on immune homeostasis and microbiota in general [137,139].

Because of its immunoregulatory functions, together with the free radical scavenging activity and anti-inflammatory properties, melatonin has also been found to be effective in combating various bacterial and viral infections [140,141]. Melatonin displayed in vitro antibacterial effects against *Staphylococcus aureus*, carbapenem-resistant *Pseudomonas aeruginosa* and *Acinetobacter baumannii*, all responsible for nosocomial infections, by reducing the availability of intracellular substrates [142]. Several studies have focused on the protection mediated by melatonin against sepsis, especially septic shock [143,144].

Melatonin also alleviates systemic inflammation caused by viruses. Melatonin’s beneficial effects have been postulated against flu infections, and also against SARS-CoV-2 virus responsible for the pandemic that has hit the world in recent months [145]. In fact, as already mentioned, melatonin reduces inflammation and oxidative stress related to aging, cardiovascular diseases, diabetes, all conditions associated with an increased risk of mortality in patients with Covid-19 disease. In addition, melatonergic pathways also appear to interact directly with viruses and have effects on influenza and Covid-19 infections. These data suggest a possible therapeutic potential of melatonin in human virus-induced disorders. In this regard, a combination of melatonin and vitamin D has been proposed as a new potential synergistic treatment, taking into account that they have many shared underlying mechanisms able to adequately modulate and control the immune system and the oxidative response against the COVID-19 infection [146].

Melatonin production is blocked by inflammatory cytokine release due to infections, and this may explain its role in regulating infective processes [147]. In addition, many of melatonin effects are obtained by optimizing the mitochondrial function. In fact, the regulation of the circadian rhythm by melatonin is mainly obtained by BMAL1 induction, the effects of which have an impact mainly on the mitochondria. Thus, virus-induced inhibition of melatonin pathways may have repercussions on circadian rhythms, and in particular on mitochondrial metabolism [147]. Supplementation with melatonin could be beneficial in restoring mitochondrial function and regulating the circadian rhythms.

Future research is needed for a better understanding of how viruses interact with exogenous melatonin and for determining the optimal therapeutic dose.

### 3.5. Antitumoral Activity of Melatonin

Immune system dysregulation, associated with a reduction of NK and T cells and an increase in TNF-α, is a common feature observed in cancer. For this reason, much attention has been paid to melatonin as therapeutic agent in addition to the classic radio/chemotherapy [148]. Several studies highlighted that melatonin mediates the initiation of immunity against tumors, even if conflicting results have been reported. Notably, a study conducted on mouse models of gastric carcinoma showed that high concentrations of melatonin were able to alter the regulatory T lymphocytes (T-reg), a subtype of CD4+ lymphocytes which are identified as CD4+CD25+ cells, being involved in tumoral cell escape from the immune system [149]. Other in vitro studies on lymphocyte cell cultures did not show this effect of melatonin on T-regs [150].

It has been hypothesized that melatonin could modulate the release of exosomes deriving from tumor cells, which are responsible for tumor progression and alteration of lymphocytes and NK cells [138]. In addition, a study showed that melatonin treatment led to a reduction in the expression of toll-like receptors (TLRs), in particular TLR4, NF-κB, IL-6 and p65 [151], but also TLR-7 and TLR-5 which are responsible for ovarian cancer invasiveness [152].

The oncostatic function of melatonin is also explained through other different strategies. In a prostate cancer study, it was hypothesized that melatonin could slow down tumor progression by counteracting the Warburg effect through a mechanism that involves entering cells through glucose transporters/solute carrier family 2A (GLUT/SLC2A) transporters, thus competing with glucose. The consequent reduction of lactate and glycolysis prevents tumor cells from obtaining energy for their survival [153]. Melatonin also modulates apoptosis by differentiating its action on the basis of cell type, either normal or cancerous. Melatonin acts as an antioxidant on normal cells by enhancing DNA repair enzymes, thus slowing down cell death and toxicity induced by radio and chemotherapy [154]; however, on most cancer cells it exerts a pro-oxidant action stimulating endogenous ROS production with consequent DNA damage and cell death [148].

Apoptosis is stimulated by increased ratio of Bax/Bcl2, but also through the modulation of p53, the main tumor suppressor protein [155]. Furthermore, melatonin is able to inhibit different antiapoptotic mediators, such as nuclear NF-κB, thus hindering Bcl2 increase but also reducing inflammatory cytokine release [156].

Recently, it has been reported that melatonin treatment of oral cancer CAL27 or SCC25 cell lines was able to inhibit the expression of non-coding micro RNAs mir-155 and mir-21, associated with poor prognosis. However, it is still necessary to address this issue in depth with long-term studies and also considering other mi-RNAs [157].

It has been observed that melatonin counteracts non-small cell lung cancer invasiveness by acting at various levels [158]. In fact, it modulates the formation of microtubules and microfilaments [159], blocks the cell cycle by delaying mitosis and phase S [160], increases the expression of occludin, a key protein of tight junctions, the downregulation of which usually promotes metastasis [161], suppresses EGFR over-expression [162], reduces Bcl-2 phosphorylation and promotes Bax [163].

Melatonin and also its metabolite AFMK, if administered with gemcitabine, were able to induce apoptosis in PANC-1 pancreatic cancer cells by modulating the Bax/Bcl-2 balance, thus representing an effective tool to improve beneficial effects of chemotherapy [164].

In addition to experimental studies indicating the potential anti-tumoral activity of melatonin through the modulation of apoptosis, autophagy and inflammation, and its enhancing action on chemotherapy beneficial effects through the reduction of side effects, preliminary clinical studies have reported beneficial effects of melatonin, alone or in combination with other therapeutic agents, in patients with gastrointestinal tumors [165].

Furthermore, given its ability to influence the synthesis of estrogens and the presence of melatonin receptors in breast tissue, melatonin has been employed in clinical studies as therapeutic agent in estrogen receptors (ER)-positive breast cancer, where it reduces the estrogenic pathway activation by down-regulating the ER receptor transcription [166]. In this regard, a greater risk of breast cancer has been associated with women who work in shifts, as exposure to light during the night inhibits the production of melatonin. Circulating melatonin and AFMK concentrations were lower in women with breast cancer undergoing chemotherapy than in healthy women, regardless of the amount of sleep [167]. In a study on female nurses, urinary a-MT6 levels appear to predict the risk of breast cancer in postmenopausal women. In particular, a lower concentration of a-MT6 has been associated with an increased risk of breast cancer and the association was not modified by tumor melatonin 1 receptor subtype [168]. However, results from these clinical studies have to be confirmed by larger prospective clinical trials.

### 3.6. Melatonin and Neurodegenerative Diseases

Melatonin metabolites AFMK, AMK and 3-OHM are known to protect brain tissues from damage through their free radical scavenging activity. Moreover, AMK is known to inhibit COX2 and neuronal NOS as well as iNOS [169]. Hence, numerous model systems have been used to characterize the mechanisms associated with melatonin deficiency in the CNS, as well as the neuroprotective effects of melatonin supplementation.

Many of melatonin effects in brain are mediated via activation of its specific receptors. Immunoreactivity analyses showed that MT1 receptor is the main melatonin receptor in human brain, being highly abundant in pituitary gland, basal forebrain, SCN, and CA3 nuclei in hippocampus, cerebellum, and retina, while MT2 receptor is mainly distributed in hippocampus and sparse in cingulate cortex, pyramidal layer, dentate gyrus, and retina [170].

Melatonin was shown to have neuroprotective effects in animal models of brain attack, i.e., by reducing the area of cerebral infarction [171,172,173]. Notably, intracerebral transplantation of pineal gland, in the presence of host intact pineal gland, protected against stroke, possibly through secretion of melatonin [174].

Evidence has been provided that melatonin is able to counteract excitotoxic cell damage. In rat cerebral cortex, melatonin was shown to inhibit calcium increase induced by acidification as well as glutamate [175,176]. Melatonin also displayed beneficial effects by partly reversing acidosis-induced abnormal dendritic complexity, synaptic protein density, imbalance of kinase/phosphatase, tau hyperphosphorylation, activation of glycogen synthase kinase-3β (GSK3β) and NF-κB pathways, endoplasmic reticulum (ER) stress and Golgi apparatus stress, and the abnormal autophagy-lysosome signals in primary cultured neurons [177].

Many observations suggest that physiological melatonin decline with aging may greatly contribute to the development of age-associated neurodegenerative disorders, such as Alzheimer’s disease (AD) and Parkinson’s disease (PD), characterized by the accumulation of protein aggregates, i.e., beta-amyloid in AD and alfa-synuclein in PD, that represent a fundamental event and a specific diagnostic marker [178].

In AD animal models, melatonin has been shown to exert a protective action against oxidative stress and cell death not only by regulating amyloid-β(Aβ)-induced alteration of calcium and mitochondrial homeostasis, but also effectively inhibiting Aβ synthesis and fibril formation. Melatonin stimulates the non-amyloidogenic processing, and down-regulates the amyloidogenic processing of Aβ precursor protein (APP), thereby preventing the formation of Aβ peptides [179]. Patients with AD exhibit lower melatonin levels than age-matched controls. In this regard, it has been hypothesized that melatonin has a protective effect on the cholinergic system by stimulating both choline transport and choline acetyltransferase (ChAT) activity and down regulating acetylcholinesterase (AChE) activity [179]. Melatonin significantly ameliorates the cognitive function in AD mice through reduction of mitochondrial damage and expression of (GSK3β), caspase-3, Aβ1-42, beta-secretase 1 (BACE1), as well as phosphorylated tau, and increase of protein phosphatase 2A (PP2A) as well as Bcl-2 [180]. In addition, melatonin is also effective against microglial activation, an important factor in AD pathogenesis, by attenuating proinflammatory cytokines and reducing oxidative damage [181].

Recently, it has been demonstrated that the treatment with melatonin or its derivatives may have beneficial effects in AD models related to tauopathy, and be effective for reducing protein aggregation and preventing cognitive decline [181]. Indeed, exogenously supplemented melatonin and AFMK were able to decrease hyperphosphorylated tau, neurofilament proteins, and malondialdehyde, increase SOD activity, improve autophagy flux for the treatment of protein aggregates, and ameliorate memory impairment, most likely through their antioxidant properties [181,182,183].

Oxidative stress plays an important role in the development of AD. The generation of free radicals caused by Aβ deposition, mitochondrial dysfunction, and inflammation is very high in AD patients. Melatonin has been reported to reduce oxidative stress-induced damage of neuronal cells and attenuate memory impairment in AD murine models [184]. The prolonged melatonin treatment in APP+/presenilin 1 (PS1) double transgenic mice reduced mitochondrial damage as well as mitophagy [185], and decreased mRNA transcripts of antioxidant enzymes SOD1, GPx, CAT [186]. Furthermore, melatonin supplement counteracts decrease in mitochondrial portion of CA1 hippocampal neurons in animal with accelerated brain aging, a phenotype similar to human geriatric disorders [187]. Melatonin supplementation significantly increased hippocampal synaptic density and abundance of excitatory synapses, decreased the number of inhibitory synapses, and up-regulated proteins synapsin I and post-synaptic density protein 95 [188]. However, meta-analysis studies suggested that melatonin supplementation in patients with either AD or dementia was able to prolong total sleep time at night, without any significant improvement of cognitive functions [189,190]. On the contrary, cognitive performance ameliorated when melatonin was added to the existing treatment for AD [191].

Using MPTP-induced PD animal models, it has been reported that melatonin injections interfere with lipoperoxidation in hippocampus and striatum, and inhibit neuronal death in nigrostriatal area [192,193]. In 6-hydroxydopamine (6-OHDA)-induced PD animal models melatonin was shown to counteract the reduction of mitochondrial oxidative phosphorylation enzyme (complex I) in substantia nigra [194], scavenge hydroxyl radical, increase GSH and cytosolic antioxidant SOD and CAT activity in neuronal cell soma and nigrostriatal pathways [195,196].

Melatonin ameliorated locomotor activity by reducing lipoperoxidation and apoptosis in paraquat-induced PD murine models [197], and by up-regulating tyrosine hydroxylase in brain striatum region as well as inhibiting striatal degeneration in a rotenone-induced PD rat model [198].

Melatonin has been reported to enhance endogenous neurogenesis through MT2 receptor activation in cerebral ischemic/reperfusion mice. The neurogenic effects of melatonin on mesenchymal stem cells were mediated by up-regulation of neurodevelopmental gene/protein expression, and reduction of oxidative/inflammatory stress, that, in turn, resulted in the preservation of blood-brain-barrier (BBB) integrity [199].

These features make melatonin a promising agent for prophylaxis and treatment of neurodegenerative disorders. Thus far, however, limited studies have evaluated the role of melatonin on clinical symptoms of neurodegenerative disorders.

A double-blind RPCT, including mild cognitive impairment (MCI) patients, showed that dietary melatonin supplementation (0.15 mg/kg for 6 months) increased the lamina cribrosa thickness (LCT) and hippocampus volume, as well as reduced CSF concentrations of T-tau, compared with placebo. Notably, the lower Mini Mental State Examination (MMSE) score, smaller hippocampus volume, and the elevated level of CSF T-tau were significantly associated with the thinner LCT in MCI patients [200]. Moreover, melatonin add-on treatment (2, 5, and 10 mg) resulted to be beneficial in MCI and AD patients by improving sleep quality and regulating the sleep/wake rhythm. However, the effects in AD patients were less pronounced [201].

A recent clinical trial showed that melatonin supplementation (10 mg/day for 12 weeks) in PD patients significantly reduced the Unified Parkinson’s Disease Rating Scale (UPDRS) part I score, Pittsburgh Sleep Quality Index (PSQI), Beck Depression Inventory (BDI), and Beck Anxiety Inventory (BAI), and ameliorated inflammatory and oxidative features together with insulin resistance [202]. Results from another trial showed that melatonin moderately improved sleep disorders in PD patients by up-regulating the expression of the clock gene BMAL1, but not PER1 [203].

It is worth highlighting some limitations of both preclinical and clinical studies on melatonin use in the treatment of neurodegenerative disorders. In particular, melatonin efficacy seems dependent on disease progression and age at which medication regimen is started. Moreover, melatonin properties in either restoring neurogenesis or modulating neurotransmission pathways involving acetylcholine and dopamine, still need to be more deeply investigated.

## 4. Contrary Data about Safety and Efficacy of Melatonin’s Therapeutic Use

Drawing final conclusions as to whether or not melatonin administration is safe may be limited by various factors, namely scarce availability of double-blind RPCT studies, weak methods used for reporting adverse effects, low statistical power unable to detect between groups differences, and lack of a priori determination of which adverse effects have to be considered relevant. Only one randomized study addressed the critical issue of melatonin administration safety as the primary outcome [204]. However, many experimental and clinical studies provided useful info about both safety and efficacy of therapeutic treatments with melatonin alone or as an add on. The short- and intermediate-term administration of melatonin produced only minor adverse effects such as agitation, dizziness, headache, nausea and sleepiness in clinical studies on children; subjective sleepiness and transient sedation, as well as a reduction—not statistically significant—of LH and GH levels in experimental studies on adults; dizziness, headache, paresthesia of mouth, arm or legs, mild headache, numbness, and worsening of dyspnea in clinical studies on adults; psychomotor impairment, sedation, disorientation, and amnesia in surgical patients, mild headache, increased sleepiness and skin rash in critically ill patients, daytime sleepiness in elderly [205].

One randomized double-blind, placebo-controlled crossover trial of long-term controlled melatonin release for the treatment of sleep disorders in children with neurodevelopmental disabilities reported mild adverse effects, namely seizures (11/51), cold/flu/infection (8/51), gastro-intestinal illness (5/51), agitation (4/51), anxiety (2/51), and headache (2/41) [206].

A recent systematic review examined 37 several studies reporting data from randomized, placebo-controlled trials involving prolonged daily melatonin administration (0.15–12 mg, for 4–29 weeks) for the treatment of primary and secondary sleep disorders. The most frequently reported melatonin adverse effects were daytime sleepiness (1.66%), dizziness (0.74%), headache (0.74%), other sleep-related adverse events (0.74%), and hypothermia (0.62%). Serious or of clinical significance adverse events were very few, including agitation, palpitations, nightmares, mood swings, fatigue, and skin irritation. Most of these effects either resolved spontaneously within a few days with no adjustment in melatonin, or immediately upon withdrawal of treatment [207].

Another challenging question is whether or not melatonin supplementation is therapeutically effective. Some meta-analysis studies found either no evidence or low to very low-quality evidence for a positive effect of melatonin supplementation on sleep disorders, cancer progression, cardio- and cerebro-vascular disorders, neurological disorders, and infertility treatment. Melatonin was not proven to be effective in treating secondary sleep disorders or sleep disorders accompanying sleep restriction, such as jet lag and shiftwork disorder, as well as consensus is lacking about outcome measures in the evaluation of melatonin efficacy for treatment of insomnia [33,208]. Additionally, a meta-analysis of randomized, controlled clinical trials, that addressed the critical issue of survival and/or tumor response after supplementation of antioxidants, including melatonin, concluded that none of the trials reported evidence of significantly increased survival time or decreased tumor growth, or both, as well as fewer toxicities than controls. However, all studies were lacking of adequate statistical power [209]. Moreover, some findings did not confirm a suppressive effect of melatonin on cancers of hematopoietic origin, and rather indicated that melatonin accelerated the proliferation of lymphoma and leukemia and restrained apoptosis of lymphoma cells [210]. No significant effects of melatonin, compared with placebo, on sleep, circadian rhythms, or agitation were observed in Alzheimer’s disease patients; moreover, sufficient scientific proof is not available for the prophylactic use of melatonin in primary headache, migraine and cluster headache [201]. Finally, a recent paper, reviewing 63 trials that involved 7760 sub-fertile women supplemented with antioxidants, including melatonin, concluded that evidence for melatonin efficacy in the treatment of subfertility was low to very low quality [211].

## 5. Conclusions

In this review, we have summarized the main biological activities of melatonin and its metabolites, underlying some healthy properties. As the deficiency of melatonin can be a marker of increased risk for disease, the determination of melatonin and its metabolites levels in body fluids may be helpful to identify subjects that need supplementation to prevent or counteract several pathological conditions.

Melatonin holds a great translational potential due to its unique antioxidant and anti-inflammatory as well as immunomodulatory properties. To our current knowledge, melatonin may be regarded as generally well tolerated and safe. Indeed, previously described mild adverse effects of melatonin treatment can be regarded as minimal if compared to short- and long-term adverse effects of some drugs, i.e., benzodiazepines, opioids, non-steroidal anti-inflammatory drugs, and glucocorticoids, that may also cause major complications and morbidities. Most importantly, melatonin treatment has no addictive properties [205].

Melatonin treatment, so far known mostly as effective in the therapy of sleep/wake cycle alterations, can be very useful in the therapeutic management of fertility disorders, osteoporosis, and oxidative/inflammatory disorders, and obesity, even if some meta-analysis studies reported conflicting results, as mentioned above. Moreover, the combination of melatonin with traditional therapies could increase the efficiency of the treatment for infective disorders, cardiovascular disease, cancer and neurodegenerative disorders, also reducing long-term side effects of conventional drugs. Indeed, the so far unestablished clinical efficacy of melatonin in many pathological settings does not allow to replace any standard therapeutic approach with melatonin treatment, except in the case of treatments of sleep disorders and as a preoperative anxiolytic [205].

Last but not least, further investigation is needed to determine the optimal dose of melatonin supplementation in humans. In this regard, critical issues are represented by difficulties in translating animal research in clinical benefit, and identifying the most effective melatonin administration way. Calculations based on animal studies showed that translational doses to humans were higher than those employed in randomized controlled trials [212]. Moreover, the bioavailability of melatonin after oral and intravenous administration in humans is very low (approximately 15%), and is influenced by age, smoking, caffeine intake, pathological conditions, specific drugs, and also food intake, especially in the case of foods containing tryptophan or serotonin [213]. Moreover, since melatonin is metabolized by cytochrome P-450 in liver, drugs competing for the same enzyme, such as fluvoxamine, caffeine, and oral contraceptives, may increase circulating levels of melatonin after exogenous melatonin administration [214].

In the light of the above-reported observations, much effort has to be spent to determine the optimal melatonin dose for long-term supplementation, and to improve supplementation ways by taking into account alternative routes, e.g., administration by intranasal, transdermal, subcutaneous, and oral transmucosal (sublingual, trans buccal) routes or administration by different preparations (elastic liposomes, spray, pastes, gels). These alternative routes are painless and allow bypassing hepatic metabolism as well as provide a sustained release increasing melatonin bioavailability [215].

## Figures and Tables

**Figure 1 antioxidants-09-01088-f001:**
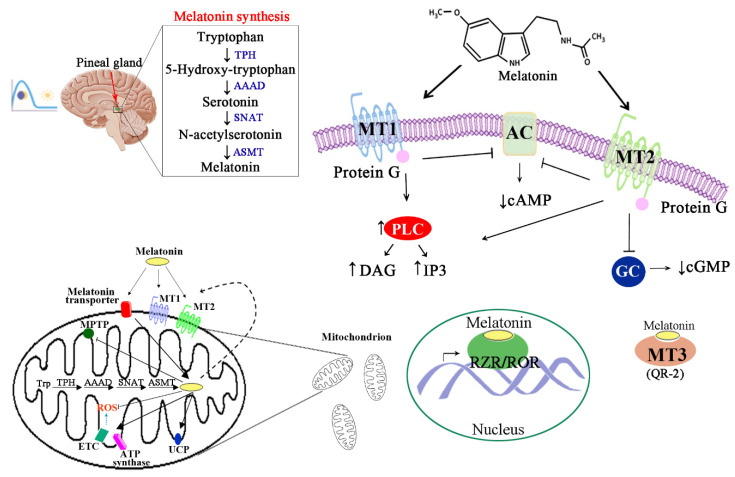
Melatonin biosynthesis and intracellular signal-transduction pathways activated by stimulation of melatonin specific receptors. The picture shows: (**figure top left**) the intracerebral site of melatonin biosynthesis, that is the pineal gland, the melatonin biosynthetic pathway, and the influence of light and photoperiod change on melatonin biosynthesis; (**figure bottom left**) mitochondrial melatonin biosynthesis and local action: melatonin is synthesized within mitochondrial matrix, but it can also enter the mitochondria through a specific transporter; melatonin activates specific MT1 and MT2 receptors on mitochondrial outer membrane, also in an autocrine way, inhibits MPTP and ROS production, and stimulates ETC and UCP; (**figure right**) melatonin interaction with its specific MT1 and MT2 receptors and RZR/ROR orphan nuclear receptors, and activation of different metabolic pathways. AAAD, aromatic L-amino acid decarboxylase; AC, adenylate cyclase; AFMK, N1-acetyl-N2-formyl-S-methoxykynuramine (melatonin metabolite); ASMT, *N*-acetylserotonin *O*-methyltransferase; cAMP, cyclic adenosine monophosphate; cGMP, cyclic guanosine monophosphate; DAG, diacylglycerol; ETC, electron transfer chain; GC, guanylate cyclase; IP3, inositol triphosphate; MPTP, mitochondrial permeability transition pore; MT1-MT3, melatonin specific receptor 1, 2, 3; PLC, phospholipase C; QR-2, quinone reductase 2; ROS, reactive oxygen species; RZR/ROR, retinoid Z receptor/retinoid acid receptor-related orphan receptor; SNAT, serotonin *N*-acetyltransferase; TPH, tryptophan hydroxylase; UCP, uncoupling protein.

**Figure 2 antioxidants-09-01088-f002:**
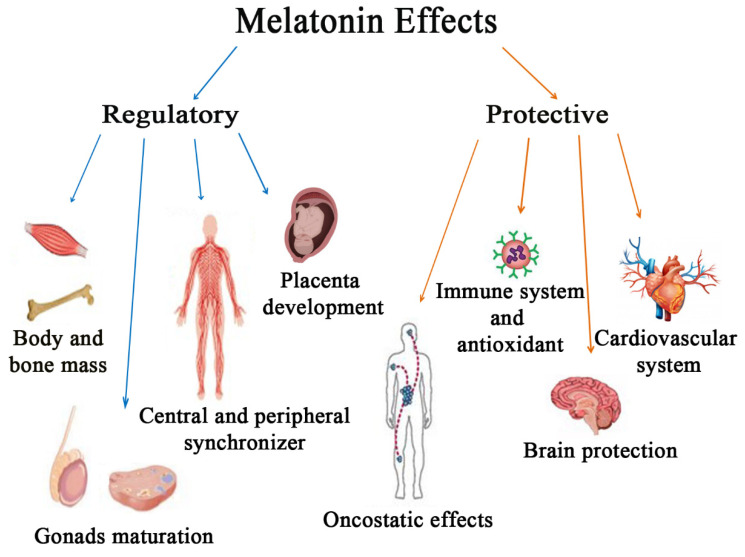
Pleiotropic actions of melatonin in human body.

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
