# Peer review of "Is Melatonin the Cornucopia of the 21st Century?"

_antioxidants, 2020, doi:10.3390/antiox9111088_

Round 1

Reviewer 1 Report

Some chapters were improved/extended substantially and I think in this form the review gained in value for the reader.
Also, some improvements were made to the figures. However, the new inclusion of a mitochondrion schematic in figure 1, originally published in https://doi.org/10.2174/1568026023394344 is not only poorly performed (very small fonts and nearly unreadable, important color info omitted) but may potentially be a case of plagiarism due to the lack of reference in the figure text. I strongly insist that this is improved for readability and correctly referenced. I suggest painting this yourself with some more connection and context to the rest of the figure... Anyway, the figure text needs to be extended in order to inform that this mitochondrial scheme shows peripheral melatonin synthesis.

90 - 112 these two paragraphs seem awfully detailed for an introduction - maybe you can shorten these paragraphs and make them easier to read

Just a few examples for typos and problematic areas (strong proofreading is still necessary)

30 ...is regulated by (the) hypothalamic suprachiasmatic nuclei ...

58 ...In the CSF the
59 concentration is higher than those (that) of the blood

67 ...so that seasonal
68 changes (in term) of temperature and photoperiod significantly affect

102 ...Notably, ROR receptors regulate(s) the biological clock

104-112 needs proofreading - wrong grammar, words wrongly placed or missing...

121 ...will not be discussed here, since

127-135 needs proofreading - wrong grammar, words wrongly placed or missing...

166 ...melatonin (Circadin) has been

318 Recently, it has been tested built a composite adhesive hydrogel system

710 ...may display toxic side effects.

Author Response

Reply to Reviewer 1

Some chapters were improved/extended substantially and I think in this form the review gained in value for the reader. 
Also, some improvements were made to the figures. However, the new inclusion of a mitochondrion schematic in figure 1, originally published in https://doi.org/10.2174/1568026023394344 is not only poorly performed (very small fonts and nearly unreadable, important color info omitted) but may potentially be a case of plagiarismdue to the lack of reference in the figure text. I strongly insist that this is improved for readability and correctly referenced. I suggest painting this yourself with some more connection and context to the rest of the figure... Anyway, the figure text needs to be extended in order to inform that this mitochondrial scheme shows peripheral melatonin synthesis.

R: Thanks for your suggestions. Accordingly, we improved figure 1 by newly painting the mitochondrion and representing mitochondrial melatonin biosynthesis and local melatonin action. Also, the text of the legend was extended in order to better detail what is represented in the picture

90 - 112 these two paragraphs seem awfully detailed for an introduction - maybe you can shorten these paragraphs and make them easier to read

R: Thanks for your comments. The mentioned paragraphs were shortened and made easier to read them.

Just a few examples for typos and problematic areas (strong proofreading is still necessary)

30 ...is regulated by (the) hypothalamic suprachiasmatic nuclei ...

58 ...In the CSF the 
59 concentration is higher than those (that) of the blood

67 ...so that seasonal 
68 changes (in term) of temperature and photoperiod significantly affect

102 ...Notably, ROR receptors regulate(s) the biological clock

104-112 needs proofreading - wrong grammar, words wrongly placed or missing...

121 ...will not be discussed here, since

127-135 needs proofreading - wrong grammar, words wrongly placed or missing...

166 ...melatonin (Circadin) has been

318 Recently, it has been tested built a composite adhesive hydrogel system

710 ...may display toxic side effects.

R: We corrected the mentioned typos and made proofreading of the mentioned paragraphs as well as other parts of the manuscript.

Reviewer 2 Report

Ferlazzo et al. Melatonin: the Cornupia of the 21st Century

The revision did not improve (see first review). Still the authors present a flattery  of the proposed effects of melatonin without mentioning studies which did not support this view.

I list here only three meta studies, which did find no or low to very low quality evidence for a positive effect of melatonin supplementation on sleep disorders, cancer progression, and infertility treatment.

  • Buscemi et al. 2006 BMJ: ´There is no evidence that melatonin is effective in treating secondary sleep disorders or sleep disorders accompanying sleep restriction, such as jet lag and shiftwork disorder. There is evidence that melatonin is safe with short term use´.
  • Block KI et al 2007 Cancer Treat Rev did a meta analysis of randomized, controlled clinical trials that reported survival and/or tumor response after antioxidant supplementation, including melatonin. They concluded: ´None of the trials reported evidence of significant decreases in efficacy from antioxidant supplementation during chemotherapy. Many of the studies indicated that antioxidant supplementation resulted in either increased survival times, increased tumor responses, or both, as well as fewer toxicities than controls; however, lack of adequate statistical power was a consistent limitation.´
  • Showell MG et al. 2020 Cochrane Database Syst Rev. 2020 analyzed the supplementation of antioxidants, including melatonin for female subfertility by reviewing 63 trials involving 7760 women. They conclude:´ In this review, there was low- to very low-quality evidence to show that taking an antioxidant may benefit subfertile women.´

Of course this is not a complete list, but clearly shows that the biased, euphemistic view of the authors do not reflect the current state of knowledge. In summary, in the present form the manuscript is not suitable for publication.

Author Response

Reply to Reviewer 2

The revision did not improve (see first review). Still the authors present a flattery  of the proposed effects of melatonin without mentioning studies which did not support this view.

I list here only three meta studies, which did find no or low to very low quality evidence for a positive effect of melatonin supplementation on sleep disorders, cancer progression, and infertility treatment.

  • Buscemi et al. 2006 BMJ: ´There is no evidence that melatonin is effective in treating secondary sleep disorders or sleep disorders accompanying sleep restriction, such as jet lag and shiftwork disorder. There is evidence that melatonin is safe with short term use´.
  • Block KI et al 2007 Cancer Treat Rev did a meta analysis of randomized, controlled clinical trials that reported survival and/or tumor response after antioxidant supplementation, including melatonin. They concluded: ´None of the trials reported evidence of significant decreases in efficacy from antioxidant supplementation during chemotherapy. Many of the studies indicated that antioxidant supplementation resulted in either increased survival times, increased tumor responses, or both, as well as fewer toxicities than controls; however, lack of adequate statistical power was a consistent limitation.´
  • Showell MG et al. 2020 Cochrane Database Syst Rev. 2020 analyzed the supplementation of antioxidants, including melatonin for female subfertility by reviewing 63 trials involving 7760 women. They conclude:´ In this review, there was low- to very low-quality evidence to show that taking an antioxidant may benefit subfertile women.´

Of course this is not a complete list, but clearly shows that the biased, euphemistic view of the authors do not reflect the current state of knowledge. In summary, in the present form the manuscript is not suitable for publication.

R: Thanks for your precious observations. In the newly revised manuscript, we have acknowledged that there are some critical issues regarding safety and efficacy of melatonin treatment, and we added a new paragraph (4) describing the main findings indicating that melatonin may have mild adverse effects and is not always effective. Accordingly, we also modified the Conclusions. Other than citing papers you mentioned, other literature references were added in the Reference list. We hope that these changes improved the quality of our manuscript.  

Reviewer 3 Report

The authors have addressed all the recommendations.

Author Response

Reply to Reviewer 3

The authors have addressed all the recommendations.

R: Thanks for your comment.

Round 2

Reviewer 2 Report

In the revision the authors now added a chapter on ´Contrary data about safety and efficacy of melatonin´s therapeutic use´, referring to the meta studies listed in the previous review and some more. Still concerning the authors maintain their euphemistic (biased) view on the efficacy of melatonin supplementation for a number of treatments (see Title and Abstract, ll 19- 23), albeit none of the meta studies find evidence for beneficial effects in treatments of secondary sleep disorders, cancer, Alzheimer´s, or infertility. The authors summarized that ´all studies were lacking of adequate statistical power´ (ll737-738). However, meta studies provide the best evidence we have, and the argument of lacking statistical power applies especially for all the pro-melatonin studies (most are non-randomized, non double-blinded, and non placebo controlled studies with small cohorts).

This conflicting presentation of hypothetical effects of melatonin supplementation needs to be addressed in the whole manuscript. The Title and the Abstract must be rephrased in a more balanced way, and the weak statistic power of single studies claiming favorable effects must be critically analyzed. This refers in particular to the chapter dealing with infertility, antitumoral activity, cardiovascular, and neurodegenerative diseases. A balanced view is expected from an authoritative review to avoid unsupported hopes of patients, as well treatments of questionable efficacy.

Author Response

Thanks for your comments and suggestions, that were very helpful in critically revising the manuscript to offer a more balanced information to the journal readership.

As you suggested, we moderately rephrased the title and the abstract. Moreover, we added new short paragraphs including comments on the need of larger randomized studies to confirm melatonin beneficial effects in the treatment of infertility, cardiovascular disorders, cancer, and neurodegenerative disorders.

In the same chapters we also added three new references and we updated the reference list with more recent publications in order to more critically address the issue of melatonin benefits.  

Round 3

Reviewer 2 Report

The authors addressed the concerns raised in the last review, now the manuscript provides a balanced discussion on melatonin effects and supplementation.

This manuscript is a resubmission of an earlier submission. The following is a list of the peer review reports and author responses from that submission.

Round 1

Reviewer 1 Report

This is a review about the main biological activities of melatonin. The authors review the effects of melatonin in cardiovascular disorders, immunomodulation, cancer, fertility, and osteoporosis, in addition to the widely known effect of sleep-wake cycle regulator. They conclude that the use of melatonin could open new opportunities for the management of several disorders.

They divided the review into ten sections. However, many sections are treated very superficially.

The authors summarize its antioxidant and anti-inflammatory. However, there are many papers describing the effects of melatonin to NFkB and NLRP3 inflammasome. The authors should discuss this point. However, thy talk about NLRP3 in the section of obesity. The review is not well organized.

Page 1, line 40: Two main enzymes control melatonin synthesis: aralkylamine N-acetyltransferase (AANAT) and acetylserotonin O-methyltransferase (ASMT).

Author Response

Thank you for your considerations. We completely revised the manuscript, by adding more details in most of the sections addressed by the review (see highlighted text) and including a description of melatonin effects to NFkB and NLRP3 inflammasome. We organize the review into four main sections, the longest of which are the second and third, respectively devoted to discussing the regulatory role of melatonin and the protective role of melatonin in different physiological and pathological settings.

We also corrected the name of the two enzymes as suggested.

Reviewer 2 Report

The review by Ferlazzo colleagues titled: "Melatoninthe cornucopia of the 21st century": addresses the wide-spread involvement of Melatonin in many if not all major human diseases. This may be due to the fact that reactive oxygen species (ROS) play an important role in many diseases and the indoleamine acts as a potent ROS scavenger and thereby alleviates oxidative stress. The review describes the main actions of melatonin on the sleep-wake cycle, as an antioxidant, its immunomodulatory, anti-infective activity, and anti-tumoral activity. Furthermore, it addresses different diseases or conditions such as obesity, cardiovascular disease, infertility, and osteoporosis. The review is well written and a good first primer to grasp the sheer broadness of melatonin's involvements besides its important physiological role as a classical circadian hormone. Due to melatonin's (and ROS's) jack-of-all-trades multifunctionality, this review uses many other reviews as a resource. However, in several parts, I missed a more thorough investigation and would have liked to see more love for detail.

Main suggestions and important improvements:
1) In reading through the different chapters I often found myself comparing the doses stated in the different experimental references (e.g. 20 mg/kg) with the initially stated physiological levels 50-100 pg/ml (page 3, line 83) and a known short-halflife of the hormone. How do these experimental treatments relate to physiological doses? Here it would be very helpful to provide more background on plasma concentrations in experimental hosts where available especially because you providing a whole chapter 2 on this topic that should cover this better. This would highly improve the comparability and give perspective in the action mode of melatonin (a pure antioxidant itself when applied in high-enough doses, or a potent messenger of antioxidative response). Finally, in the abstract, you mention melatonin supplementation but in the text, there is a total lack of further information on this... what are common doses, long-term effects, etc... Please address these points.

2) The two figures are of poor quality and need a thorough redesign. For example in Fig. 1 please use the same abbreviations for enzymes as used in the text and figure text (e.g. NQO2 vs. QR-2). Please streamline this. Font sizes are arbitrary and using consistent font size (and caps) information and readability could be improved. The same applies to the use of colors, thick or thin lines, big and small arrows, etc... Also, the light blue nucleus in the lower-left corner is too small in relation to and considering its important role in the circadian clock. Please add information that ROR is the entry point of melatonin into the circadian clock. The melatonin molecule structure is too thin and small in relation to other fonts and lines. In several instances you write that melatonin is mainly produced in the mitochondria - this would be interesting to add in this figure. Stretching the font vertically should be avoided. Consider using a simple sans-serif font. In figure 2 all words are underlined - this makes no sense.

3) I like the grouping of Melatonin effects into "regulatory" and "protective" effects in figure 2. This could be also applied when structuring the text into chapters. For me, it was not always clear what's disease, what's physiological function of melatonin. Maybe you can relate this grouping also to neuroendocrine hormone response and autocrine local antioxidant function of melatonin.

4) Given this overview of Fig. 2, I miss a chapter on brain protection - please extend your short notion at line 196 and add a chapter on Melatonin and neurodegenerative diseases. I think this would improve your review considerably.

5) Please clarify the sites of melatonin synthesis in more detail. There is the pineal gland and there are extra-pineal sites, and then there are mitochondria - I am confused - this could be described with better detail and give many readers new ideas since I think this is quite an unknown territory. Please clarify this part of the introduction and maybe extend figure 1.

6) The conclusion is very short. I would have liked to see more aspects that talk against the notion of melatonin as a "cornucopia". Could you add some critical aspects maybe? One aspect to me is the physiological relevance of many of the in vitro experiments (see again point 1 above). Also, it is known from other hormones and vitamin supplements that long-term use can become detrimental and that reactive oxygen species can have hormetic effects on the organism.

Errors and minor issues:
abstract:
line 11: pinealocytes
line 23: I miss the neurodegenerative diseases also here.

line 30: ...have also shown...
36: ...inhibited by light...
39: transferase
50: what does NAT stand for - please use same abbreviations for the different enzymes
60: add space after the alpha
62: add short information on the circadian clock signaling pathway as important recipient of ROR signaling
73:...in body fluids...
93: Could you add half a sentence on the nature of such limitations?
102: ...dependent on central nervous...
160: ...C3-OHM is...
193: ...hepato-pulmonary...
230: ...the best-known inflammasome...
237: ...a preadipocyte cell line...
238: ...decreasing phosphorylated... ...signal-regulated kinase (p-ERK)...
245: ...the regulation of production (production of what?)
255: ...of other adipokine levels...
258-263: This part is hard to read/understand - simplify
277: ... melatonin may play a key role.
280: The sentence "Melatonin is, therefore..." is to my opinion weakly placed - move it further down (e.g. line 293) and combine the two parts to one paragraph since this is still the same topic.
283: ... pineal gland of a fetus...
289-293: rephrase and simplify
307: ...NAD(+) and...
310: ...model of heart infarct...
335: You talk about seasonal changes of melatonin levels - this could also be mentioned in more detail in chapter 2.
348: PBMCs needs to be defined once
403: add paragraph after ...survival [110].
408: remove paragraph after ...cell death [112].
420: ...a protein of tight...
422: [121], and reduces...
490: This is one example where the reader could benefit from actual doses instead of "high doses" - see my point 1) above
538: ...with no side effects...

Author Response

The review by Ferlazzo colleagues titled: "Melatoninthe cornucopia of the 21st century": addresses the wide-spread involvement of Melatonin in many if not all major human diseases. This may be due to the fact that reactive oxygen species (ROS) play an important role in many diseases and the indoleamine acts as a potent ROS scavenger and thereby alleviates oxidative stress. The review describes the main actions of melatonin on the sleep-wake cycle, as an antioxidant, its immunomodulatory, anti-infective activity, and anti-tumoral activity. Furthermore, it addresses different diseases or conditions such as obesity, cardiovascular disease, infertility, and osteoporosis. The review is well written and a good first primer to grasp the sheer broadness of melatonin's involvements besides its important physiological role as a classical circadian hormone. Due to melatonin's (and ROS's) jack-of-all-trades multifunctionality, this review uses many other reviews as a resource. However, in several parts, I missed a more thorough investigation and would have liked to see more love for detail.

Main suggestions and important improvements:
1) In reading through the different chapters I often found myself comparing the doses stated in the different experimental references (e.g. 20 mg/kg) with the initially stated physiological levels 50-100 pg/ml (page 3, line 83) and a known short-halflife of the hormone. How do these experimental treatments relate to physiological doses? Here it would be very helpful to provide more background on plasma concentrations in experimental hosts where available especially because you providing a whole chapter 2 on this topic that should cover this better. This would highly improve the comparability and give perspective in the action mode of melatonin (a pure antioxidant itself when applied in high-enough doses, or a potent messenger of antioxidative response). Finally, in the abstract, you mention melatonin supplementation but in the text, there is a total lack of further information on this... what are common doses, long-term effects, etc... Please address these points.

R: Thank you for your considerations. The manuscript has been totally revised by adding more details in most of the sections addressed (see highlighted text) covering all the points you suggested. This is an important issue. An orally administrated dose of 0.3mg melatonin is considered physiological and raise plasma melatonin to levels within the normal nocturnal range (i.e.60–200 pg/ml).

As observed in the study of Zhdanova and collaborators (J Clin Endocrinol Metab. 2001) the median peak levels of melatonin, observed on average within 2 h of treatment, were 84 (59–120) g/ml, 220 (124–299) pg/ml, or 1370 (957–2440) pg/ml after oral administration of the 0.1-, 0.3-, or 3.0-mg doses, respectively.

To concerns animal studies, we hopeful these have taken into account the rules about translation of doses between species (A simple practice guide for dose conversion between animals and human J Basic Clin Pharm. March 2016-May 2016; 7(2): 27–31.)

2) The two figures are of poor quality and need a thorough redesign. For example in Fig. 1 please use the same abbreviations for enzymes as used in the text and figure text (e.g. NQO2 vs. QR-2). Please streamline this. Font sizes are arbitrary and using consistent font size (and caps) information and readability could be improved. The same applies to the use of colors, thick or thin lines, big and small arrows, etc... Also, the light blue nucleus in the lower-left corner is too small in relation to and considering its important role in the circadian clock. Please add information that ROR is the entry point of melatonin into the circadian clock. The melatonin molecule structure is too thin and small in relation to other fonts and lines. In several instances you write that melatonin is mainly produced in the mitochondria - this would be interesting to add in this figure. Stretching the font vertically should be avoided. Consider using a simple sans-serif font. In figure 2 all words are underlined - this makes no sense.

R: In the new version of the manuscript figures have been changed according to your suggestions, however we prefer not to include mitochondria in the figure, a sentence in the main text (line…) has been added.

3) I like the grouping of Melatonin effects into "regulatory" and "protective" effects in figure 2. This could be also applied when structuring the text into chapters. For me, it was not always clear what's disease, what's physiological function of melatonin. Maybe you can relate this grouping also to neuroendocrine hormone response and autocrine local antioxidant function of melatonin.

R: In the new version of the manuscript we addressed this point by organizing the review into four main sections, the longest of which is the second and third, respectively devoted to discussing the regulatory role of melatonin and the protective role of melatonin in different physiological and pathological settings.

4) Given this overview of Fig. 2, I miss a chapter on brain protection - please extend your short notion at line 196 and add a chapter on Melatonin and neurodegenerative diseases. I think this would improve your review considerably.

R: A new chapter on Melatonin and neurodegenerative diseases has been added in the new version of the manuscript.

5) Please clarify the sites of melatonin synthesis in more detail. There is the pineal gland and there are extra-pineal sites, and then there are mitochondria - I am confused - this could be described with better detail and give many readers new ideas since I think this is quite an unknown territory. Please clarify this part of the introduction and maybe extend figure 1.

R: In accordance with your comment, In the newly revised manuscript version we better describe the synthesis of melatonin and modified Figure 1.

6) The conclusion is very short. I would have liked to see more aspects that talk against the notion of melatonin as a "cornucopia". Could you add some critical aspects maybe? One aspect to me is the physiological relevance of many of the in vitro experiments (see again point 1 above). Also, it is known from other hormones and vitamin supplements that long-term use can become detrimental and that reactive oxygen species can have hormetic effects on the organism.

R: Thank you for the comment. The Conclusions were extended. Moreover, we tried to improve the criticism you mentioned. Nevertheless, about in vitro experiments, it is known the difficulty of extrapolating from experimental concentration to physiological level of melatonin, as well as comparing in vitro concentrations with in vivo dose. However, one has to take into account that in vitro researches are mostly aimed to investigate the mechanism of action of the molecule and cellular response.

We also added some sentences on the reported few adverse effects of melatonin long-term administration, that have been so far observed in studies focusing on the treatment of sleep disorders. However, literature data on hermetic effects of melatonin are completely missing.

Errors and minor issues:
abstract:
line 11: pinealocytes
line 23: I miss the neurodegenerative diseases also here.

line 30: ...have also shown...
36: ...inhibited by light...
39: transferase
50: what does NAT stand for - please use same abbreviations for the different enzymes
60: add space after the alpha
62: add short information on the circadian clock signaling pathway as important recipient of ROR signaling
73:...in body fluids...
93: Could you add half a sentence on the nature of such limitations?
102: ...dependent on central nervous...
160: ...C3-OHM is...
193: ...hepato-pulmonary...
230: ...the best-known inflammasome...
237: ...a preadipocyte cell line...
238: ...decreasing phosphorylated... ...signal-regulated kinase (p-ERK)...
245: ...the regulation of production (production of what?)
255: ...of other adipokine levels...
258-263: This part is hard to read/understand - simplify
277: ... melatonin may play a key role.
280: The sentence "Melatonin is, therefore..." is to my opinion weakly placed - move it further down (e.g. line 293) and combine the two parts to one paragraph since this is still the same topic.
283: ... pineal gland of a fetus...
289-293: rephrase and simplify
307: ...NAD(+) and...
310: ...model of heart infarct...
335: You talk about seasonal changes of melatonin levels - this could also be mentioned in more detail in chapter 2.
348: PBMCs needs to be defined once
403: add paragraph after ...survival [110].
408: remove paragraph after ...cell death [112].
420: ...a protein of tight...
422: [121], and reduces...
490: This is one example where the reader could benefit from actual doses instead of "high doses" - see my point 1) above
538: ...with no side effects...

R: We corrected all the reported errors and made the suggested modifications in the paragraphs.

Reviewer 3 Report

Ferlazzo et al Melatonin: the cornucopia of the 21th century

In this review the authors present a flattery of the effects of melatonin (supplementation) on human health and welfare, cumulating in the last sentence of the Abstract ´Melatonin supplementation holds a great beneficial potential … to the management of oxidative stress- and/or inflammation-related disorders, such as obesity, cardiovascular diseases, immune disorders, infectious diseases, cancer as well as osteoporosis and infertility.´ However, in stark contrast they conclude in last sentence oft he Conclusions `… however, it appears a safe molecule, with no mild effects (sic), more clinical trials, especially long-term studies, are necessary to clarify the effects of melatonin on human health.´

So what?

In summary, this is not a systematic and critical review of the state of the art in melatonin research, but an un-critical listing of proposed and hypothetical mechanisms. In particular, papers, which do not describe positive effects, seem to be neglected, e.g. Buscani et al. BMJ 2006. As a reader I m disappointed after the lecture, since this draft do not provide me with any meaningful summary, concepts or questions. In short, I do not recommendate this work for publication.

Author Response

In the newly revised version the manuscript was organized into four main sections, the longest of which are the second and third, respectively devoted to discuss the regulatory role of melatonin and the protective role of melatonin in different physiological and pathological settings.

Moreover, we added few paragraphs, throughout the text and in section Conclusions, describing the few reported adverse aeffects of melatonin treatment.

We did not find the paper you mentioned by Buscani et al. BMJ 2006. Instead, we found a paper by Buscemi et al. BMJ 2006 that states melatonin is safe and don’t show any adverse effects.